# Indonesia-Based Study of the Clinical and Cost-Saving Benefits of Subcutaneous Allergen Immunotherapy for Children with Allergic Rhinitis in Private Practice

**DOI:** 10.3390/cells10071841

**Published:** 2021-07-20

**Authors:** Anang Endaryanto, Ricardo Adrian Nugraha

**Affiliations:** 1Department of Child Health, Faculty of Medicine, Universitas Airlangga—Dr. Soetomo General Hospital, Surabaya 60235, Indonesia; 2Department of Cardiology and Vascular Medicine, Faculty of Medicine, Universitas Airlangga, Surabaya 60235, Indonesia; ricardo.adrian.nugraha-2019@fk.unair.ac.id

**Keywords:** allergic rhinitis, subcutaneous immunotherapy, clinical benefit, cost-saving benefit

## Abstract

Background: Until now, the cost of allergy treatment in the insured public health care system and the non-insured self-financing private health care system in Indonesia has not been well documented and published, as well as the cost of allergy treatment with subcutaneous immunotherapy. Objective: To evaluate the clinical and cost benefits of allergic rhinitis treatment in children with subcutaneous immunotherapy in a non-insured self-financing private health care system. Methods: A retrospective cohort study conducted from 2015 until 2020 that compared the clinical improvement and health care costs over 18 months in newly diagnosed AR children who received SCIT versus matched AR control subjects who did not receive SCIT, with each group consisting of 1098 subjects. Results: A decrease in sp-HDM-IgE level (kU/mL) from 20.5 + 8.75 kU/mL to 12.1 + 3.07 kU/mL was observed in the SCIT group. To reduce the symptom score of allergic rhinitis by 1.0 with SCIT, it costs IDR 21,753,062.7 per child, and for non-SCIT, it costs IDR 104,147,878.0 per child. Meanwhile, to reduce the medication score (MS) by 1.0 with SCIT, it costs IDR 17,024,138.8, while with non-SCIT, it costs IDR 104,147,878.0. Meanwhile, to lower combination symptoms and medication score (CSMS) by 1.0, with SCIT, it costs IDR 9,550,126.6, while with non-SCIT, it costs IDR 52,073,938.9. Conclusions: In conclusion, this first Indonesia-based study demonstrates substantial health care cost savings associated with SCIT for children with AR in an uninsured private health care system and provides strong evidence for the clinical benefits and cost-savings benefits of AR treatment in children.

## 1. Introduction

Allergic rhinitis (AR) is one of the most important diseases of the natural course of allergic disease [1]. There is a tendency for less-than-optimal AR treatment by health workers [2]. AR that is not well controlled will burden patients, nurses, employers, and the health care system [3,4]. Children with uncontrolled AR have a greater risk of developing asthma, recurrent otitis media, chronic rhinosinusitis, and other comorbid conditions [5,6]. Children with AR also face a decreased quality of life, which manifests as sleep disturbances, poor school performance, decreased energy, a depressed mood, and low frustration tolerance [7,8]. The economic costs of poorly controlled AR include: the cost of non-prescription drugs to treat symptoms, prescription drug costs, and the cost of medical care for comorbid complications, such as asthma and acute sinusitis [9]. Indirect costs of uncontrolled AR include absenteeism from school, decreased productivity of children and their parents, loss of parents’ daily wages, and injuries resulting from fatigue [3,9,10]. In 2011, the estimated total US direct AR costs exceeded USD14 billion, with 60% of the spending on prescription drugs [11]. In Indonesia, the highest prevalence of allergies in big cities is AR, followed by asthma and atopic dermatitis [12,13,14,15]. The majority of AR patients in Indonesia are school-age children [12,14,15]. The main cause of AR in children in Indonesia is house dust mite allergen [12,14,15]. AR sufferers in Indonesia are dominated by school age children [12,14,15]. AR that is not managed properly will lead to asthma and disrupt children’s quality of life [3,9]. Allergen Immunotherapy (AIT) is an effective and safe therapeutic method for curing AR. AIT is an underutilized treatment for AR both in the USA and elsewhere in the world including Indonesia. Only 3–5% of US children and adults with AR, asthma, or both have received AIT [16].

Until now, the cost of allergy treatment in the insured public health care system and the non-insured self-financing private health care system in Indonesia has not been well documented and published, as well as the cost of allergy treatment with AIT. It has not been required to apply AIT as an allergy treatment option in the public or private health care system in Indonesia. The AIT commonly used in Indonesia is subcutaneous immunotherapy (SCIT). For the public health care system in Indonesia, AIT with the SCIT method has been routinely carried out in Dr. Soetomo General Academic Hospital, Surabaya, and the costs can be covered by health insurance organized by the Indonesian government. For the private health care system in Indonesia, there is only one private allergy clinic that provides special services for allergic children that is equipped with SCIT services and accepts special referrals for allergic children who require SCIT from general practitioners, paediatrician, and other specialists from all areas in Indonesia. We conducted an Indonesian-based study with the aim of evaluating the clinical and cost benefits of AR treatment in children with SCIT in a non-insured self-financing private health care system.

## 2. Materials and Methods

### 2.1. Ethical Approval

This observational research procedure has received approval from the Health Research Ethics Committee of Dr. Soetomo General Academic Hospital, Surabaya, Indonesia (2021), with reference number 0297/LOE/301.4.2/I/2021. Subjects and their parents were briefed on the study before they agreed to provide clinical data and cost-of-treatment data for this observational study. Implied consent was obtained from parents and caregivers. General data—including name, address, age, gender, weight, height, and telephone number—were collected and recorded for all subjects. Likewise, specific data, for example, comorbid allergic asthma and its duration, as well as details about other allergies and their medications, were also recorded.

### 2.2. Subjects

Subjects in this study were children diagnosed with allergic rhinitis. Subjects’ inclusion criteria included children aged 3–18 years with complete clinical and cost-of-treatment data [16]. The diagnosis and treatment of AR and asthma are consistent with allergic rhinitis and its impact on asthma (ARIA) [17] and the New Global Initiative for Asthma (GINA) [18] and results of reactive skin tests using HDM allergens. Exclusion criteria for subjects included abnormal shape in the anatomy of the nose and paranasal sinuses, patients diagnosed with cancer, autoimmune disease, cerebral palsy, and Down syndrome. Parents or legal guardians first received an explanation of the purpose of the study before they agreed that their child’s clinical data belonged to us as our study data. They must complete and sign the consent form if they agree to provide clinical data and cost-of-treatment data for this observational study.

### 2.3. Study Design

A retrospective cohort study (2015–2020) compared the clinical improvement and health care costs over 18 months in newly diagnosed AR children who received SCIT versus matched AR control subjects who did not receive SCIT. The SCIT group and the non-SCIT group consisted of 1098 subjects each (Figure 1). The SCIT group was the group that received AR therapy plus SCIT HDM, while the non-SCIT group only received AR therapy. The AR therapy given includes antihistamines, intranasal steroids, and systemic steroids, and subjects will be given bronchodilators, skin care, and physiotherapy according to the symptoms of other accompanying allergic diseases [17,19]. The study examined data from Children’s Allergy Consultant Private Clinic in Surabaya, Indonesia, from 1 January 2015 to 31 March 2020. HDM SCIT was given once a week for 14 weeks and then continued once every 3 weeks until the 18th month. Data on weight, height, family allergy history, symptom score, treatment score, symptom-treatment combination score, and comorbid atopic conditions (asthma, conjunctivitis, and atopic dermatitis) before and after presenting SCIT were recorded. Data on the frequency of consultations with physicians (family physicians and specialists), hospitalizations (including emergencies), and rehabilitation care or physiotherapy during the 18 months of SCIT indicated. Likewise, data regarding the use of resources and funding for laboratory examinations, consultations with physicians (family physicians and specialists), prescribed drugs, emergency care, outpatient care, hospitalization, and travel costs for physician and medical examinations were documented for 18 months. observation. All clinical data and cost-of-treatment data from patients and physicians were documented by the research team into the study database.

### 2.4. Study Procedure

SCIT is given to patients whose symptoms have not been controlled by medical treatment and symptomatic therapy. In accordance with the Indonesian Paediatric Society’s treatment strategy, SCIT therapy can be given to patients whose symptoms cannot be controlled with normal allergy medical treatment for at least 3 months. Subjects included AR patients who experienced wheezing, coughing, and/or shortness of breath. In subjects with asthma, spirometry tests were performed. After proper treatment and their asthma was well controlled as well as normal spirometry tests, SCIT was started. Adverse reactions to SCIT are classified into 2 categories: local reactions and systemic reactions. Local reactions were defined as erythema, pruritus, and swelling at the injection site. A systemic reaction was defined as a life-threatening range from mild to very severe anaphylaxis [20]. The house dust mite allergen immunotherapy (Teaching Industry Allergen by Dr. Soetomo Hospital-Airlangga University, Surabaya, Indonesia) used was an extract of *Dermatophagoides pteronyssinus* with 11.3–26.6 ng/mL via subcutaneous injection [21,22,23]. The dose of immunotherapy used every week varies: 0.1 cc (first week), 0.15 cc (second week), 0.22 cc (third week), 0.32 cc (fourth week), 0.48 cc (week fifth), 0.72 cc (sixth week), 1 cc (seventh week), 0.1 cc (eighth week), 0.15 cc (ninth week), 0.22 cc (tenth week), 0.32 cc (eleventh week), 0.48 cc (twelfth week), 0.72 cc (thirteenth week), and 1 cc (fourteenth week) [22]. The use of HDM immunotherapy is based on previous studies in Indonesia that stated that the most common types of HDM were *Dermatophagoides pteronyssinus* (87%), *Dermatophagoides farinae* (7%), and *Bromia tropicalis* (6%) [24]. Another study also states that the most common HDM found in Indonesia is *Dermatophagoides pteronyssinus,* which can be found in various places such as beds, floors, and sofas, while *Dermatophagoides farinae* is most commonly found on sofas. *Bromia tropicalis* was the least compared to *Dermatophagoides pteronyssinus* and *Dermatophagoides farinae* [25].

### 2.5. Outcomes

#### 2.5.1. Allergic Rhinitis Symptoms

AR and asthma symptoms were monitored according to allergic rhinitis and its impact on asthma (ARIA) [17] and the Global Initiative for Asthma (GINA) guidelines [18]. Signs and symptoms of AR include itchy nose, sneezing, rhinorrhea, or nasal congestion, as well as itching of the roof of the mouth, post-nasal drip, and cough.

#### 2.5.2. Measurement of SPT (Skin Prick Test) and Serum Specific IgE Levels

The skin prick test (SPT) was applied to all patients to check their sensitivity to the following HDM allergens (Allergopharma, Reinbek, Germany). Their reactivity to HDM allergens in SPT was identified through allergen-induced wheals equal to or larger in diameter than histamine-induced wheals. The measurement of serum-specific IgE levels used an enzyme-linked immunosorbent assay/ELISA (Euroline TM; Euroimmun AG, Lübeck, Germany). Not all subjects were examined for spHDM IgE levels but were randomly selected by 10% of the total sample. An indirect ELISA examination was conducted by taking up to 6 mL of the subject’s blood serum from the vein, which was then rotated at 3000 rpm for 15 min. The serum was stored at −20 °C to keep the condition stable. Specific ring allergens for certain allergens were inserted in the wells and then incubated with the patient’s sample. Should the patient’s sample be positive, the specific IgE in the subject’s serum would bind to the allergen. This antibody–gene binding could be detected by adding a monoclonal anti-human IgE conjugate. The length of time to check serum IgE levels was 3.5 h. The serum IgE determination was in the range of 0.35–100 kU/L.

#### 2.5.3. Measurement of Symptom Scores and Drug Scores

The measurement of symptom scores, drug scores, and symptom and drug combined scores is a form of clinical evaluation in addition to measuring SPT for household dust mites and specific IgE (for HDM). Clinical evaluations were carried out during the study period, which began on 1 January 2014 and ended on 31 March 2020. Subject diary cards recorded symptoms (nose, eyes, mouth, and lungs) and treatment scores (antihistamines, local steroid drugs, systemic steroids), including symptoms in the nose (sneezing, stuffiness and runny nose), eyes (itching, redness, tears and swelling), mouth and throat (itching and dryness), and chest (shortness of breath, coughing, wheezing, and tightness), on a scale of 0–3 (with a score of ‘0’ indicating no symptoms and ‘1,’ ‘2,’ and ‘3’ showing mild, moderate, and severe symptoms, respectively). The daily treatment score is scored based on the type and amount of rescue medication used each day. In addition, each dose of decongestant, antihistamine, and inhaled corticosteroid used was marked as 1 [26]. Asthma symptoms were recorded daily for the entire treatment period on the scorecard. Daytime symptom scores included: 0—no symptoms; 1—wheezing; 2—temporary asthma attacks; 3—permanent asthma. Overnight symptom score included: 0—no symptoms; 1—wheezing; 2—1–2 asthma attacks; 3—more than 3 asthma attacks. Any additional medication required for the treatment of asthma was also recorded on a daily basis [27]. The level of asthma control was recorded in each patient’s standard hospital record according to the GINA guidelines. Subject diary cards recorded symptom scores and treatment scores, including daytime symptoms, activity limitations, night time symptoms or awakening, need for relief or rescue medication, lung function, and history of exacerbations. All patients were evaluated on symptom scores, drug scores, and symptom and drug combined scores per period (trimester 1, trimester 2, semester 2, semester 3) after starting the observation.

#### 2.5.4. Measurement of AR and SCIT Treatment Costs

The medical costs associated with each treatment are calculated as the total costs associated with the following categories of resource use: all-cause admissions, SCIT-related outpatient visits, and non-SCIT-related outpatient visits. The total cost per period (trimester 1, trimester 2, semester 2, semester 3) is calculated based on the cost per unit usage and data usage accordingly. The AR care expense calculated here is due to medical costs (e.g., medical services used for treatment) and travel costs for travel for physician and medical examinations. Direct non-medical costs (e.g., caregiver costs and household modifications), as well as indirect costs of disability, early retirement, and parent’s job loss due to childcare activities, were not documented in this study. The cost is calculated by multiplying the frequency of diagnostic and treatment services by the cost of each activity. The unit cost at the 2020 price is used to estimate the use of health care resources to determine the cost of managing patients in Indonesia in both groups (SCIT and non-SCIT). The cost for private outpatient services for a specialist or family doctor is IDR 150,000. The prescribed medication is documented from patient records. Hospitalization costs are calculated using the daily average rate for accommodation (IDR 850,000), medical services (IDR 200,000), and rehabilitation therapy (IDR 775,000). The following direct costs are noted, based on clinical studies: medications (antihistamines, corticosteroids, and bronchodilators), skin care, physiotherapy, allergen immunotherapy, and hospital health care. The time horizons in the cost analysis were set to 3, 6, 9, and 18 months (trimester 1, trimester 2, semester 2, semester 3). The reason is that the SCIT outcome will provide valuable information after 3, 6, 9, and 18 months. Cost per patient for drugs is based on prices from pharmacies in Indonesia and is calculated assuming the intake of antihistamines based on cetirizine (0.25 mg/kg/dose every 12–24 h and a price of IDR 14,730 per 5 mg/mL 60 mL bottle syrup), pseudo ephedrine (0.25–1.0 mg/kg/dose every 3 to 4 h and priced at IDR 77,000 per 60 mL bottle), intranasal corticosteroids (based on fluticasone or mometasone, 50 mcg, 2 puffs/day, and a price of IDR 240,000 per puff package), fluticasone/mometasone-based inhaled corticosteroid, 125 mcg, 2 puffs/day, and a price of IDR 225,000 per 120 puff package), a salbutamol-based bronchodilator (0.1 mg/kg/dose every 8 h and a price of IDR 120,521 per 2 mg/5 mL bottle syrup 60 mL), oral corticosteroid based on oral methylprednisolone (0.5–1.0 mg/kg/dose every 12 to 24 h with a maximum dose; 60 mg/4 h and a price of IDR 7163 per 8 mg tablet). Treatment of oral H1 antihistamines and intranasal corticosteroids were prescribed to all patients for a minimum of 3 months. Oral antihistamines H1 and intranasal corticosteroids are prescribed for AR, and inhaled corticosteroids are prescribed for patients with asthma. Laboratory costs that are calculated specifically are costs for diagnosing allergic rhinitis and its comorbidities with allergic causes, including costs for radiological examinations, SPT and HDM-specific IgE. Travel costs for travel for physician and medical examinations are based on geographic location and the distance between the patient’s residence and the doctor’s practice. This travel cost calculation uses the standard cost of taxis, flights, hotels, and other travel costs adjusted to the rates that apply at the location where the patient lives. The unit cost at the 2020 price is used to estimate the use of travel costs.

### 2.6. Statistical Analysis

A sample size of 2196 patients provided sufficient strength (90%) with a Minimum Detectable Effect (MDE) of 0.125 between the 1098 subjects in the SCIT group and the 1098 subjects in the control group (*p* = 0.01). Each patient in the group treated with SCIT matched at least 1 patient in the control group on all 8 of the following variables: age group, gender, physician specialist at referral, family allergy history, symptom score, treatment score at the initiation of SCIT, and comorbid atopic conditions (asthma, conjunctivitis, and atopic dermatitis) as long as they are worn before the initiation of SCIT. Control patients who were matched for these variables must also have at least 18 months of data after their match date. If patients treated with SCIT could not be matched on all 8 variables for at least 1 control patient, then patients treated with SCIT were excluded from further analysis. The data were first tested using Kolmogorov–Smirnov. Furthermore, data on participant’s characteristics were analysed using independent *t*-test or the Mann—Whitney test. In addition, other measurement results were analysed using an independent *t*-test or Mann—Whitney test and a dependent *t*-test or Wilcoxon test. The statistical test results were declared significant if *p* < 0.05. Data analysis used IBM SPSS Statistics software version 23.0 (IBM Corp., Armonk, NY, USA).

## 3. Results

### 3.1. Characteristics of Study Participant

Figure 1 displays the results of the sample identification procedures. Among all Paediatric Allergy Immunology Consultant’ patients enrolees (*n* = 7356), among whom there were 83.7% (6126/7356) received a diagnosis of AR; among the 6126 enrolees with newly diagnosed AR, 47.8% (2920) received de novo SCIT. Overall, there were 1797 SCIT-treated patients and 2313 control subjects eligible for matching; from this pool, 1098 SCIT-treated patients were matched to 1098 control subjects.

The distribution of the children with allergic rhinitis that were recruited as subjects of this study based on geographic area in Indonesia, as shown in Figure 2. A total of 2196 children with allergic rhinitis from 92 Districts in the Republic of Indonesia were recruited as subjects of this study, they came from (District and number of children): Surabaya 879, Sidoarjo 425, Jombang 135, Kupang 95, Lamongan 77, Gresik 40, Kediri 31, Pamekasan 31, Sampang 31, Jember 30, Sumenep 29, Mataram 28, Lumajang 27, Tulungagung 24, Mojokerto 21, Tuban 20, Balikpapan 18, Pasuruan 17, Bangkalan 16, Bojonegoro 14, Madiun 12, Nganjuk 12, Samarinda 12, Probolinggo 11, Denpasar 9, Lumajang 9, Sampit 6, Banjarmasin 5, Banyuwangi 5, Kertosono 5, Magetan 5, Malang 5, Palangkaraya 5, Pamekasan 5, Bondowoso 4, Nabire 4, Situbondo 4, Sorong 4, Mataram 3, Martapura 3, Maumere 3, Merauke 3, Muaratewe 3, Porong 3, Tarakan 3, Trenggalek 3, Waingapu 3, Bandung 2, Batulicin 2, Blitar 2, Cepu 2, Jakarta 2, Jayapura 2, Manado 2, Mojokerto 2, Palu 2, Pandaan 2, Pangkalan Boen 2, Semarang 2, Sumbawa 2, Alor 1, Ambon 1, Bandar Lampung 1, Bangil 1, Batu 1, Baubau 1, Berau 1, Besuki 1, Bima 1, Blega 1, Blora 1, Ende 1, Flores 1, Jambi 1, Kavamenano 1, Kotabaru 1, Kutai 1, Lawang 1, Makasar 1, Palembang 1, Parigi 1, Prigen 1, Sangata 1, Saumlaku 1, Sepanjang 1, Singaraja 1, Soe 1, Solo 1, Tanah Grogot 1, Timika 1, Timor 1, Tobelo 1, and Wonosobo 1. Subjects come from almost all islands in the Republic of Indonesia, consisting of (Name of Island or Islands and Number of children): Java 1956, islands in Nusa Tenggara 142, Kalimantan 68, islands in Papua 13, Bali 10, islands in Maluku 3, Sulawesi 2, and Sumatra 2.

Table 1 shows stratification according to sex, age, body weight/height, comorbid illness, and illness severity. The 1098 children in the SCIT-treated matched sample were predominantly male (63.4%); the mean age at the initial AR diagnosis was 5.5 (SD 3.51) years; mean body weight at the initial AR diagnosis was 12.8 (SD, 2.35) kilograms; the mean body height at the initial AR diagnosis was 84.7 (SD, 19.83) centimetres. In the year before their initial AR diagnoses, the majority (87.3%) of these children experienced one comorbid disease burden, and the rates of: asthma, atopic dermatitis, sinusitis, and conjunctivitis were 54.0%, 9.1%, 1.1%, and 0.3%, respectively. In total, 88.6% of our subjects were previously treated by a paediatrician, whilst those previously treated by family doctors, ENT specialists, and dermatologists, were 2.5%, 2.6%, and 6.3%, respectively. The majority of severity at the initial AR diagnosis was moderate, and the rate of the severity of mild, moderate, and severe were 46.6%, 46,6%, and 6.7%, respectively. Children originated equally from East Java Region 1 (Surabaya) 43.9%, East Java Region 5 (Bojonegoro) 37.1%, East Java Region 4 (Jember) 4.2%, East Java Region 3 (Madiun) 2.8%, and East Java Region 2 (Malang) 1.3%, as well as from outer Province (0.4%) and outer Java Island (10.1%). As shown in Table 1 compared with matched control subjects, children who subsequently received SCIT experienced significantly more overall comorbid disease burden in the year before SCIT initiation. Whereas asthma and atopic eczema occurred significantly less frequently among matched controls, rates of other diseases of the upper respiratory tract, such as sinusitis and other respiratory system diseases, such as bronchitis, were significantly higher among SCIT-treated patients.

### 3.2. Symptom Scores, Drug Scores, SPT of House-Dust-Mite Diameter, IgE-Specific House-Dust-Mite

Table 2 shows the effect of SCIT on symptom scores, drug scores, SPT of house-dust-mite diameters, and IgE-specific house-dust-mites. The mean symptom score (SS) of the children in the first trimester (0–3 months) in the SCIT group was 2.4 (SD 0.61), similar to that in the non-SCIT group. Likewise, the medication score (MS) and combination symptom and medication score (CSMS) are the same in the first trimester. The decline in SS to 1.7 in 4–6 months, to 0.8 in 7–12 months, and a slight increase to 1.2 in 13–18 months occurred with SCIT. Meanwhile, in the non-SCIT group, scores did not decrease at 4–6 months, fell slightly to 2.2 at 7–12 months, and to 2.0 at 13–18 months. There was a significant difference in SS reduction with SCIT from 4–6 months to 13–18 months. The reduction in SS score on SCIT was 1.8 (SD 0.45) significantly different (*p* = 0.000) from the reduction in SS score on non-SCIT (0.4 (SD 0.48)).

The mean medication score (MS) of the first trimester (0–3 months) in the SCIT group was 2.6 (SD 0.48), the same as in the non-SCIT group. The decline in MS to 1.7 at 4–6 months, to 0.8 at 7–12 months, and to 0.4 at 13–18 months occurred in the SCIT group. Meanwhile, for non-SCIT, the MS score decreased to 2.4 at 4–6 months, to 2.2 at 7–12 months and remained 2.2 at 13–18 months. There was a significant difference in the decline in MS at SCIT from 4–6 months to 13–18 months. The reduction in MS score on SCIT was 2.3 (SD 0.59) significantly different (*p* = 0.000) from the reduction in MS score on non-SCIT (0.4 (SD 0.49)). The mean combination symptom and medication score (CSMS) in the first trimester (0–3 months) in the SCIT group was 5.0 (SD 0.76), similar to that in the non-SCIT group. The decrease in CSMS to 3.5 in 4–6 months, to 1.6 in 7–12 months, and to 1.5 in 13–18 months occurred in the SCIT group. Meanwhile, for non-SCIT, the CSMS score decreased to 4.7 in 4–6 months, to 4.4 at 7–12 months, and to 4.2 at 13–18 months. There was a significant difference in the decrease in CSMS at SCIT from 4–6 months to 13–18 months. The reduction of the CSMS score on SCIT was 4.1 (SD 0.80) and significantly different (*p* = 0.000) from the reduction of the CSMS score on non-SCIT (0.8 (SD 0.84)).

The mean SPT mite diameter (mm) and sp-HDM-IgE level (kU/mL) in the SCIT group were 9.3 (SD 4.17) mm and 20.5 (SD 8.75) kU/mL, respectively, which were not significantly different from the non-SCIT group at the beginning of observation (0 months). The decrease in SPT mite diameter (mm) from 9.3 (SD 4.17) mm at the beginning of observation (0 months) to 6.2 (SD 1.14) mm at the end of the observation (18 months) occurred in the SCIT group. Meanwhile, in the non-SCIT group, there was also a decrease in SPT Mite diameter (mm) from 9.3 (SD 4.15) mm at the beginning of the observation to 7.6 (SD 3.91) mm at the end of the observation (18 months). There was a significant difference in SPT Mite diameter (mm) between the SCIT and non-SCIT groups after 18 months of observation. The reduction diameter of SPT Mite (mm) on SCIT was 2.4 (SD 1.26) and significantly different (*p* = 0.000) from the reduction diameter of SPT Mite (mm) in non-SCIT (1.7 (SD 0.71)). 

The decrease in sp-HDM-IgE level (kU/mL) from 20.5 (SD 8.75) kU/mL at baseline of observation (0 months) to 12.1 (SD 3.07) kU/mL at the end of observation (18 months) occurred in the SCIT group. Meanwhile, in the non-SCIT group, there was also a decrease in the sp-HDM-IgE level (kU/mL) from 20.3 (SD 8.66) kU/mL at the beginning of the observation to 16.4 (SD 9.58) kU/mL at the end of the observation (18 months). There was a significant difference in sp-HDM-IgE levels (kU/mL) between the SCIT and non-SCIT groups after 18 months of observation. Reduction level of sp-HDM-IgE (kU/mL) on SCIT was 8.4 (SD 8.93) kU/mL significantly different (*p* = 0.000) with reduction of sp-HDM-IgE level (kU/mL) in non-SCIT (3.9 (SD 1.73)).

Table 3 shows each effect of SCIT on points of symptoms (nasal, eye, and lung symptoms). 

### 3.3. Frequencies of Resources Utilization

Table 4 shows resource utilization, such as physician and medical prescription. The average number of times SCIT group children visited a physician in 0–3 months (the first trimester), 4–6 months (second trimester), 7–12 months (second semester), 13–18 months (third semester), and in total, 18 months was 9.1 (SD 2.69) times, 9.1 (SD 0.50) times, 8.1 (SD 0.44) times, 3.5 (SD 1.18) times, and 29.8 (SD 3.34) times, respectively, and at the same time, the average number of medical prescriptions required was 29.7 (SD 10.87) times, 15.9 (SD 5.26) times, 6.7 (SD 4.96) times, 1.4 (SD 3.02) times, and 53.7 (SD 17.79) times, respectively.

The average number of times non-SCIT group children visited a physician in 0–3 months (the first trimester), 4–6 months (second trimester), 7–12 months (second semester), 13–18 months (third semester), and in total, 18 months was 9.5 (SD 2.42) times, 8.2 (SD 1.99) times, 7.6 (SD 1.64) times, 6.1 (SD 1.39) times, and 31.4 (SD 7.07) times, respectively, and at the same time, the average number of medical prescriptions required was 30.0 (SD 10.41) times, 19.8 (SD 4.86) times, 17.1 (SD 5.73) times, 14.0 (SD 5.46) times, and 80.9 (SD 20.70) times, respectively.

The average number of times children visited a physician in 4–6 months (second trimester) and 7–12 months (second semester) in the SCIT group were higher than the non-SCIT group but, in the past 13–18 months (third semester), were lower than the non-SCIT group. The average number of medical prescriptions required in 0–3 months (the first trimester) was not significantly different between the SCIT group and the non-SCIT group. However, from since second trimester (4–6 months), the average number of medical prescriptions in the SCIT group were lower than the non-SCIT group.

The most common agents of medical prescriptions used by AR patients were antihistamine/decongestant combinations w/or w/o analgesic or cough suppressant 42.8% (7731 times), followed by local (nasal) corticosteroids, w/or w/o inhaled corticosteroids 33.4% (6032 times); systemic corticosteroids 14.7% (2663 times); inhaled long-acting β2 agonists, w/or w/o short-acting β2 agonists 7.6% (1379 times); and emollient 1.4% (259 times). Drug agents were used more by the non-SCIT group (11,251 times) than by the SCIT group (6828 times). AR patients were mainly treated with oral antihistamines. Patients suffering from both illnesses often received intranasal agents (e.g., beclomethasone dipropionate or disodium cromoglycate) and inhaled disodium cromoglycate/nedocromil. The use of prescribed inhaled sympathomimetics and corticosteroids increased proportionately to illness severity.

Children in the SCIT group decreased the average number of medication episodes over 18 months from 1.0 to 0.66 (34.0%) for antihistamine/decongestant combinations w/or w/o analgesic or cough; from 1.0 to 0.44 (56.3%) for local (nasal) corticosteroids, w/or w/o inhaled corticosteroids; from 0.63 to 0.19 (70.4%) for systemic corticosteroids; from 0.45 to 0.01 (87.3%) for inhaled long-acting β_2_ agonists, w/or w/o short-acting β_2_ agonists; from 0.08 to 0.01 (87.0%) for emollient. Children in the non-SCIT decreased the average number of medication episodes over 18 months from 1.0 to 1.0 (0.0%) for antihistamine/decongestant combinations w/or w/o analgesic or cough; from 1.0 to 0.84 (15.7%) for local (nasal) corticosteroids, w/or w/o inhaled corticosteroids; from 0.63 to 0.19 (70.4%) for systemic corticosteroids; from 0.36 to 0.11 (69.2%) for inhaled long-acting β_2_; from 0.08 to 0.02 (73.0%) for emollient.

A few hospitalizations occurred during the study. Overall, 1.3% (*n* = 30) were treated as inpatients, for an average of 2.3 days (SD 0.83) in the SCIT Group (*n* = 14) and for an average of 3.1 days (SD 1.91) in the non-SCIT Group (*n* = 16). In terms of the number of inpatient stays over the 18-month period, SCIT-treated children had significantly fewer inpatient stays than their matched counterparts. A few hospitalizations occurred during the study. Overall, 1.3% (*n* = 30) were treated as inpatients, for an average of 2.3 days (SD 0.83) in the SCIT group (*n* = 14) and for an average of 3.1 days (SD 1.91) in the non-SCIT group (*n* = 16). In terms of the number of inpatient stays over the 18-month period, SCIT-treated children had significantly fewer inpatient stays than their matched counterparts. The average number of hospitalizations required in all periods of observations was not significantly different between the SCIT group and the non-SCIT group.

### 3.4. Medical Costs

Table 5 shows pharmacotherapy and physiotherapy costs. For antihistamine/decongestant combinations w/or w/o analgesic or cough, the total average cost was IDR 3,061,626.6 (SD 1,490,628.18) in the SCIT group and IDR 33,699.2 (SD 1,566,519.59) in the non-SCIT group. For local (nasal) corticosteroids, w/or w/o inhaled corticosteroids, the total average cost was IDR 7,543,623.5 (SD 4504589.31) in the SCIT group and IDR 11,432,952.5 (SD 5,268,534.95) in the non-SCIT group. For systemic corticosteroids, the total average cost was IDR 941,928.0 (SD 919,279.26) in the SCIT group and IDR 1,787,334.1 (SD 1,700,252.88) in the non-SCIT group. For inhaled long-acting β2 agonists, w/or w/o short-acting β_2_ agonists, the total average cost was IDR 262,012.8 (SD 284,388.49) in the SCIT group and IDR 271,779.07 (SD 444,767.99) in the non-SCIT group.

In the SCIT group, the average costs of antihistamine/decongestant combinations w/or w/o analgesic or cough suppressant in 0–3 months (the first trimester) and 4–6 months (second trimester) but, from 7–12 months (second semester), was lower than the non-SCIT group. The average costs of systemic corticosteroids and skin care in 0–3 months were higher than the non-SCIT group but from 4–6 months (second trimester) was lower than the non-SCIT group. The average costs of local (nasal) corticosteroids, w/or w/o inhaled corticosteroids, inhaled long-acting β2 agonists, w/or w/o short-acting β2 agonists, in the SCIT group were not different with the non-SCIT group but after 4–6 months (second trimester) until 13–18 months (third semester) were lower than the non-SCIT group.

Physiotherapy was prescribed for 7.4% (*n* = 163) of all patients. In the SCIT group, the rate of rehabilitation prescription in 0–3 months (the first trimester), 4–6 months (second trimester), 7–12 months (second semester), and 13–18 months (third semester) was 1.1%, 0.1%, 0.1%, and 0.1%, respectively. In the non-SCIT group, the rate of rehabilitation prescriptions in 0–3 months (the first trimester), 4–6 months (second trimester), 7–12 months (second semester), and 13–18 months (third semester) was 9.7%, 1.3%, 1.3%, and 1.3%, respectively. For skin care (emollient), the total average cost was IDR 129,207.7 (SD 487,025.11) in the SCIT group and IDR 150, 831.5 (SD 652,406.98) in the non-SCIT group. The average costs of physiotherapy in the 18-month period evaluation were lower than the non-SCIT group. The mean total cost for all types of pharmacotherapies given to AR children in this study during the 18 months of observation in the SCIT group was lower than that in the non-SCIT group. SCIT-treated children incurred significantly lower total pharmacotherapy costs than their matched counterparts.

The average of total pharmaco-physiotherapy costs in 0–3 months (IDR 6,396,033.2 (SD IDR 3,483,707.81)) in the SCIT group was no different from the non-SCIT group (IDR 6,449,759.4 (SD IDR 33,103,405.56)), but from 4–6 months (second trimester), the average of total pharmaco-physiotherapy in the SCIT group was lower than the non-SCIT group. The average of total pharmaco-physiotherapy given to AR children in this study during the 18 months of observation in the SCIT group (IDR 12,037,214.6 (SD IDR 6,699,040.21) was lower than that in the non-SCIT group (IDR 17,935,392.6 (SD IDR 7,661,506.27)). The mean 18-month per-patient cost savings for outpatient visits achieved by the SCIT group was almost 1.5 times greater (IDR 12,037,215 vs. IDR 17,935,393 *p* < 0.0001) than that achieved by the non-SCIT group compared with matched control subjects, patients who received SCIT incurred significantly lower mean per-patient total health care costs within 3 months of treatment initiation; this significant effect persisted over the 18-month follow-up period. Significant differences (*p* < 0.0001) in mean total health care costs occurred at 3-, 6-, 12-, and 18-month follow-up.

In the SCIT group, the average of physician costs in 0–3 months (the first trimester) and in 13–18 months (third semester) were lower than the non-SCIT group, but in 4–6 months (second trimester) and in 7–12 months (second semester), the average was higher. The average of physician costs in this study during the 18 months of observation in the SCIT group was lower than that in the non-SCIT group.

The average of laboratory costs in 0–3 months (the first trimester) in the SCIT group, was no different from the non-SCIT group, but in 13–18 months (third semester), it was higher. The average of laboratory costs in this study during the 18 months of observation in the SCIT group (IDR 6,491,463.5 (SD IDR 204,254.75)) was a little bit higher (1002 more higher) than in the non-SCIT group (IDR 6,501,985.8 (SD IDR 186,589.33)). By adding travel costs to the total pharmaco-physiotherapy costs, the total medical costs were obtained. The average of medical costs in 0–3 months (the first trimester) in the SCIT group was higher than in the non-SCIT group, and in 4–6 months (second trimester), it was not different from the non-SCIT. From 7–12 months (second semester), the total medical costs in the SCIT group were lower. The average of medical costs in this study during the 18 months of observation in the SCIT group (IDR 26,574,651.3 (SD 6,926,884.04)) was lower than in the non-SCIT group (IDR 28,548,577.4 (SD IDR 8,099,994.40)).

By adding travel costs to total medical costs, as seen in Table 6, we get total health care costs. The average of total health care costs in 0–3 months (the first trimester) in the SCIT group was higher than in the non-SCIT group. From 4–6 months (second trimester), the total health care costs in the SCIT group were lower. The average of total health care costs in this study during the 18 months of observation in the SCIT group (39,155,519.2 (SD 7,671,303.08)) was 0,93 times lower than that in the non-SCIT group (IDR 41,659,151.1 (SD 10,031,917.63)). However, in the non-SCIT group, the symptom score (SS) was still 1.4 times higher than in the SCIT group, the medication score (MS) was still 5.5 times higher than in the SCIT group, and the combination symptoms and medication score (CSMS) was still 2.8 times higher than in the SCIT group.

For children with AR, to reduce the symptom score (SS) by 1.0 with SCIT, it costs IDR 21,753,062.7 per child, and for non-SCIT, it costs IDR 104,147,878.0 per child. Meanwhile, to reduce the medication score (MS) by 1.0 with SCIT, it costs IDR 17,024,138.8, while with non-SCIT, it costs IDR 104,147,878.0. Meanwhile, to lower the combination symptoms and medication score (CSMS) by 1.0 with SCIT, it costs IDR 9,550,126.6, while with non-SCIT, it costs IDR 52,073,938.9.

## 4. Discussion

In this retrospective, matched cohort study, we compared matched groups of children with newly diagnosed AR who either did or did not receive SCIT after their first AR diagnosis. Even after matching groups by age, sex, body weight, body height, family history of allergy, symptom and medication scores at SCIT initiation, and comorbid atopic conditions (asthma, conjunctivitis, and atopic dermatitis) during the year prior to SCIT initiation, SCIT-treated children incurred 32.9% (IDR 12,037,215 vs. IDR 17,935,393 *p* < 0.0001) lower mean 18-month drugs costs, 31.6% (IDR 33,470 vs. IDR 48,907 *p* < 0.0001) lower mean 18-month total inpatient costs, and 9.5% (IDR 39,155,519.2 vs. IDR 41,659,151.1 *p* < 0.0001) lower mean 18-month total health care costs, respectively, after SCIT initiation. Furthermore, these significant reductions were evident as early as 3 months after immunotherapy initiation and increased during the 18-month follow-up period. Our study demonstrated savings in reducing symptom scores (SS), medication scores (MS), and combination symptoms and medication scores (CSMS). To reduce SS by 1.0, SCIT saved IDR 82,394,815.3 per child (IDR 21,753,062.7 per child with SCIT vs. IDR 104,147,878.0 per child without SCIT). Meanwhile, to reduce MS by 1.0, SCIT saved IDR 87,123,739.2 per child (IDR 17,024,138.8 per child with SCIT vs. IDR 104,147,878.0 per child without SCIT). Meanwhile, to reduce CSMS by 1.0, SCIT saved IDR 42,523,812.3 per child (IDR 9,550,126.6 per child with SCIT vs. IDR 52,073,938.9 per child without SCIT).

Another study [28,29] found that the cost of immunotherapy was offset by the cost savings gained 3 months [28] and 6 months [29] after the completion of immunotherapy. Our study found that SCIT costs could be offset by total cost savings in the 2nd semester (SCIT cost IDR 1,200,000.0 vs. total-health-care-cost savings of IDR. 2,788,540.0 per child), and in the 3rd semester (SCIT cost IDR 300,000.0 vs. total-health-care-cost savings of IDR 1,973,926.1 per child). An Italian retrospective study [30] compared the average direct health care costs of 135 children and adolescents with AR and asthma, asthma alone, or AR, asthma, and conjunctivitis during the year before starting immunotherapy with those obtained during 3 years of immunotherapy. Compared to the year prior to the initiation of immunotherapy, the average annual total healthcare costs per patient were 56% lower over the 3 years of immunotherapy. The investigators also found no significant difference in the mean direct healthcare costs per annual patient (>4 years) for the subset of asthmatic patients who had received immunotherapy (*n* = 41) and a matched sample of asthma patients who had not received immunotherapy. Our study found that compared to the first semester of SCIT initiation, the average total health care cost per patient in semester 3 was 73% lower during 18 months of SCIT, while those who were not immunotherapy were only 52% lower. Ariano et al. [31] conducted a 6-year prospective study in which 30 patients with seasonal rhinitis and asthma were randomly assigned to receive 3 years of immunotherapy or pharmacological treatment and then followed up for an additional 3 years after completion of treatment. Although no significant cost difference was seen in the first year of treatment, patients treated with immunotherapy had 15% (*p* < 0.001) and 48% (*p* < 0.001) lower health care costs in the second and third years of treatment, respectively. In our study, a significant cost difference was seen in the second and third semesters of treatment, and patients treated with SCIT had 29% (*p* < 0.001) and 36% (*p* < 0.001) lower health care costs in the second semester, and to the third semester, respectively. In Ariano et al.’s [31] study, this statistically significant cost difference was maintained for 3 years after discontinuation of immunotherapy and peaked at 80% (*p* < 0.001 in the sixth year of the study (the third year after discontinuation of immunotherapy). The mean annual net savings the average over 6 years was USD830. In addition to these retrospective and prospective cost studies, several European economic modelling studies have provided support for the cost-effectiveness of immunotherapy [32].

In the most interesting study demonstrating the cost-effectiveness of AIT, Ariano et al. [33] examined the pharmacoeconomics of SCIT with SDT versus standard therapy alone among 30 patients with at least 2 years of history of AR and Parietaria-induced asthma. Patients were treated for 3 years and followed for an additional 3 years after discontinuation of treatment. In addition to tracking symptom scores and drug use, the researchers calculated the costs of patients’ health care, including scheduled and unscheduled clinic visits and prescription medications. In the first year of the study, those receiving SCIT experienced significant improvement in symptoms and reduced drug use compared with patients receiving standard therapy alone. In the second year of treatment, SCIT provided a cost savings of 48% compared to standard therapy alone. At the end of the study, the investigators reported that SCIT provided an annual net savings per patient of USD 830, representing an 80% cost reduction compared to the standard therapy group without SCIT. In our study, since the first 3 months, those who received SCIT experienced significant improvement in symptoms and reduced drug use (Table 2) compared to patients who received standard therapy alone. Our study found that compared to the first semester of SCIT initiation, the average total health care cost per patient in semester 3 was 73% lower during 18 months of immunotherapy, while those who were not immunotherapy were only 52% lower. At the end of the study (3rd year) in Ariano et al.’s study [31], the investigators reported that SCIT provided annual net savings per patient of USD830, representing an 80% cost reduction compared to the standard therapy group without SCIT, whereas, at the end of the in our study (18 months), SCIT has provided annual net savings of IDR 4,136,199.2 (equivalent to USD289) per patient representing a 73% cost reduction. Similar to Ariano et al.’s study [31], our study was also designed to cover not only the direct costs of SCIT but also the total health care costs (i.e., all prescribed medications, scheduled and unscheduled medical clinic visits, and so on), but with subjects, our study was much larger (*n* = 2198 AR children).

Another study of savings due to SCIT can be seen in a 2-year double-blind placebo-controlled trial (DBPC) comparing SCIT-administered ragweed allergen extract with a placebo for the treatment of 77 adolescents and adults with ragweed-induced asthma. The investigators reported significant improvement in clinical and economic outcomes in the patient population receiving SCIT compared to the placebo group [32]. The costs of asthma medications and allergen extracts used during the 2-year study period were considered. Over the course of the study (24 months), the cost of asthma treatment for those receiving SCIT was USD 840 versus USD 1194 for those receiving placebo; this represents the 30% cost savings benefit provided by SCIT. However, these cost savings were offset by the USD 527 spent on supplies and SCIT-related administrative costs. During our study (18 months), the cost of treating allergic rhinitis for those receiving SCIT was IDR 12,037,214 (equivalent to USD 842) versus IDR 17,935,392.6 (equivalent to USD 1255) for those receiving non-SCIT; this represents the 32% cost savings benefit provided by SCIT. The cost savings in our study were also offset by IDR 4,200,000 (equivalent to USD 293) spent on supplies and SCIT-related administrative costs. A 2005 Danish study examining the direct and indirect costs of AIT for seasonal grass pollen allergy and house dust mite (HDM) allergy patients revealed significant savings associated with AIT [34]. Prior to the initiation of AIT, the direct annual cost per patient for allergy treatment is 2580 Danish Krone (equivalent to IDR 6,044,802.6 or USD 422). In the years following discontinuation, the direct annual cost per patient fell to DKK 1072 (equivalent to IDR 2,518,407.1 or USD 176), representing 60% savings. When direct and indirect costs are considered, the annual cost per patient is significantly less with SCIT than standard therapy without SCIT. This study further supports the cost-saving benefits of SCIT for patients with allergic respiratory conditions. In a claims-based analysis of Florida Medicaid patients, Hankin and colleagues [28] compared the direct costs (pharmaceutical, outpatient and inpatient services for any reason) incurred by paediatric patients newly diagnosed with AR in the 6 months prior to the initiation of SCIT with direct costs. This patient parallel occurred within 6 months of discontinuation of SCIT. The investigators found a significant reduction in costs (*p* < 0.001) in the 6-month period following SCIT, even after SCIT costs were included.

In an extension of the previously mentioned data, Hankin and colleagues [29] performed a retrospective, matched cohort analysis of 10 years of claims data (1997–2007) to examine whether children with newly diagnosed AR who received SCIT expended less health care utilization and fewer costs over the 18-month follow-up period compared to the group of AR-matched children who did not receive SCIT. Children treated with SCIT incur significantly lower total health care costs of 18 months per patient even after including allergen immunotherapy (IT)-related costs (USD 3247 vs. USD 4872), outpatient costs excluding SCIT-related care (USD 1107 vs. USD 2266), and pharmaceutical costs (USD 1108 vs. USD 1316) compared with matched control subjects (*p* < 0.001 for all). In our study children treated with SCIT incur lower mean total health care costs of 18 months per patient even after including costs associated with SCIT (IDR 39,155,519.2, which equates to USD 2739 vs. IDR 41,659,151.1, which equates to USD 2908), costs outpatient care excluding SCIT-related care (IDR 34,922,049.3, which equates to USD 2438 vs. IDR 41,610,244.0, which equates to USD 2905), and pharmaceutical costs (IDR 12,037,214.6, which equates to USD 842 vs. IDR 17,935,392.6, which equates to USD 1255) compared to matched control subjects (*p* < 0.001 for all). Consistent with previous studies, a significant difference in Cost-Effectiveness of SCIT in total health care costs was evident in the first 3 months of starting immunotherapy in our study. Our study shows that the initial cost savings associated with SCIT persist and, more importantly, increase over the 18-month study period.

In a retrospective large-scale, matched cohort claims analysis evaluating 12 years of Florida Medicaid data, Hankin and colleagues [3] found comparable cost savings in adults with newly diagnosed AR treated with SCIT. In this study, researchers compared the average 18-month health care costs (pharmaceutical, outpatient and inpatient) of adult and paediatric patients with newly diagnosed AR who received SCIT with those who did not. Specifically, SCIT treatment was associated with savings in children of a 30% reduction in total 18-month health care costs (USD 5253 SCIT vs. matched control subjects USD 9118; *p* < 0.0001). These savings in health care costs, including costs associated with SCIT, were evident in the first 3 months of treatment and continued throughout the 18 months of follow-up. In contrast to Hankin et al.’s study, which analysed claims from Medicaid insurance subjects, our study analysed the costs incurred by patients from private practice with non-insurance self-financing. In our study, when we compared the average 18-month health care costs (after we added pharmaceutical, outpatient and inpatient costs, and of course no travel costs), SCIT treatment was associated with savings (in paediatric AR patients) of 7% reduction in total 18-month health care costs (IDR 26,574,651.3 or USD 1855 SCIT vs. matched control subjects IDR 28,548,577.4 or USD 1992; *p* < 0.0001). The overall cost savings (7%) shown in our research seem low, but the progress of savings increased significantly when viewed from the progress in the 3-month, 6-month, 12-month, and 18-month period, which was −19%, respectively, −3%, 29%, and 36%.

From a systematic review conducted by the National Institute for Health Research Health Technology Assessment (HTA) program identifying 14 economic evaluations and two economic evaluation reviews [35]. It was concluded that on a GBP 20,000–30,000 per QALY basis, SCIT is cost-effective compared to therapeutics. The standard course is 6 years of initiation of SCIT treatment. From a National Health Service perspective, SCIT is cost-effective after 7 years, and SCIT is also found to be cost-effective compared to SLIT after 5 years. Limited evidence suggests SCIT may be more useful and less expensive than SLIT. The authors note that the studies used different outcome measures making it difficult to compare and combine results. Our study did not evaluate beyond 18 months, but at the end of 18 months, SCIT was superior to non-SCIT in reducing symptom scores (SS), medication scores (MS), and combination symptom and medication scores (CSMS). To reduce SS by 1.0, SCIT saved IDR 82,394,815.3 per child, to reduce MS by 1.0, SCIT saved IDR 87,123,739.2, henceforth, to reduce CSMS by 1.0, SCIT saved IDR 42,523,812.3 per child. 

### Strength and Limitations

Several limitations should be mentioned about this study, although we have attempted to match patients with potentially confounding variables but may have been unable to control for other important characteristics, such as patient adherence to pharmacological treatment and allergen avoidance. Allergen avoidance in the allergy guidelines of the Indonesian Paediatric Society is recommended as the first step in allergy treatment, but there is no guarantee that all families will comply, so we cannot determine whether SCIT and non-SCIT matched patients are equally likely to comply with instructions on allergen avoidance. In addition, we also do not have information regarding the implementation of the avoidance measures that we recommend to his parents. Because our study subjects were enrolled in care through a non-insured self-financing private health care system, these findings may not be generalizable to individuals receiving care through an insurance-financed public health care system. Although several studies, including this study, have found that SCIT-related cost savings increase over time, the duration of follow-up (18 months) was generally limited by subjects dropping out of SCIT (not following up until 18 months after SCIT initiation).

This is the first study in Indonesia to show a significant comparison of AR patients on standard therapy with SCIT with patients on standard therapy without SCIT as a matched control in health care costs as early as 3 months after initiation of treatment. With these results, it is hoped that the use of SCIT is more frequent in Indonesia because SCIT is proven to not only improve clinical outcomes but also reduce medical expenses early and consistently in children with AR.

## 5. Conclusions

In conclusion, this first Indonesia-based study demonstrates substantial health care cost savings associated with SCIT for children with AR in an uninsured private health care system and provides strong evidence for the clinical benefits and cost-savings benefits of AR treatment in children.

### Patient and Public Involvement Statement

How was the development of the research question and outcome measures informed by patients’ priorities, experience, and preferences? Yes, parents and advisers had been informed about research questions and outcomes measured.How did you involve patients in the design of this study? Patients were not involved.Were patients involved in the recruitment to and conduct of the study? Patients were not involved.How will the results be disseminated to study subjects? The results will be disseminated to study subjects and parents.

## Figures and Tables

**Figure 1 cells-10-01841-f001:**
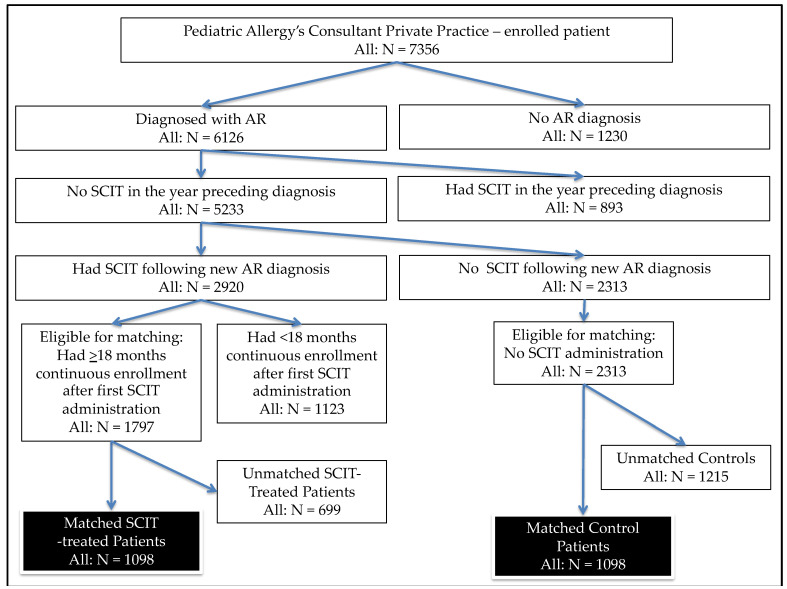
The identification of matched samples. Note: The 8 matching variables were: age, sex, body weight, body height, family history of allergy, symptoms, and medication scores at the SCIT initiation and comorbid atopic conditions (asthma, conjunctivitis, and atopic dermatitis) during the year prior to the SCIT initiation.

**Figure 2 cells-10-01841-f002:**
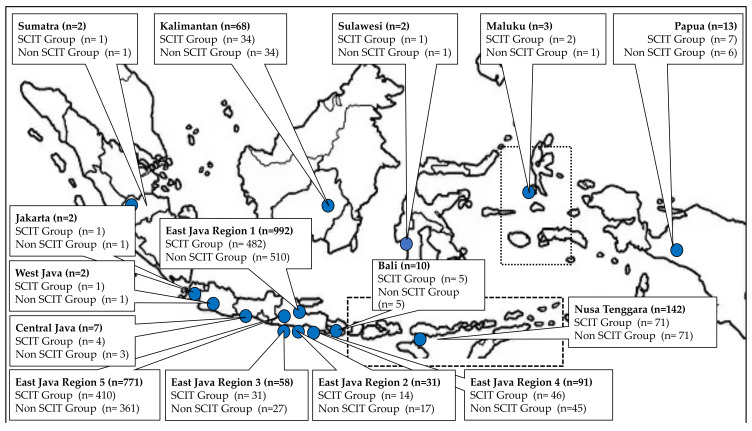
The distribution of the 2196 children with allergic rhinitis that were recruited as subjects of this study based on geographic area in Indonesia (Island, Province or Area).

**Table 1 cells-10-01841-t001:** Characteristics Subjects.

Characteristics	SCIT Group (*n* = 1.098)				Control Group (*n* = 1.098)			*p*
Age category (years), *n* (%)								
0–1	44		(4.0)		48		(4.4)	0.199
2–5	542		(49.4)		538		(49.0)	
6–12	432		(39.3)		458		(41.7)	
13–18	54		(4.9)		36		(3.3)	
18+	26		(2.4)		18		(1.6)	
Sex, *n* (%)								
Male	696		(63.4)		704		(64.1)	0.722
Female	402		(36.6)		394		(35.9)	
AR—associated conditions, *n* (%)								
Asthma, *n* (%)	593		(54.0)		396		(36.1)	0.000
Bronchitis, *n* (%)	431		(39.3)		700		(63.8)	0.000
Atopic dermatitis, *n* (%)	100		(9.1)		85		(7.7)	0.249
Sinusitis, *n* (%)	12		(1.1)		106		(9.7)	0.000
Conjunctivitis, *n* (%)	3		(0.3)		0		(0.0)	0.083
GI Problem, *n* (%)	7		(0.6)		3		(0.3)	0.205
Urticaria, *n* (%)	14		(1.3)		20		(1.8)	0.300
Nutrition Status								
BW (kgs), mean (SD)	12.8	(2.35)		12.7		(2.33)		0.761
BH (cm), mean (SD)	84.7	(19.84)		84.2		(19.24)		0.530
% BW/Age	82.3	(3.29)		82.3		(3.26)		0.938
% BH/Age	77.9	(4.11)		77.9		(4.07)		0.938
Number of comorbidity, *n* (%)								
0	40	(3.6)		11		(1.0)		0.000
1	959	(87.3)		870		(79.2)		
2	96	(8.7)		211		(19.2)		
3	3	(0.3)		6		(0.5)		
Physician specialty of referral, *n* (%)								
Primary care/general practitioner	27	(2.5)		27		(2.5)		0.277
Paediatrician	973	(88.6)		952		(86.7)		
ENT specialists	69	(6.3)		74		(6.7)		
Dermatologist	29	(2.6)		45		(4.1)		
Geographic region, *n* (%)								
East Java Region 5 (Bojonegoro)	410	(37.3)		361		(32.9)		0.325
East Java Region 4 (Jember)	46	(4.2)		45		(4.1)		
East Java Region 3 (Madiun)	31	(2.8)		27		(2.5)		
East Java Region 2 (Malang)	14	(1.3)		17		(1.5)		
East Java Region 1 (Surabaya)	482	(43.9)		510		(46.4)		
Outer Province	4	(0.4)		7		(0.6)		
Outer Island	111	(10.1)		131		(11.9)		
Baseline Symptom Score (SS), mean (SD)	2.7	(0.48)		2.7		(0.45)		0.398
Baseline Skin Prick Test Diameter (Mite) (mm), mean (SD)	9.3	(4.17)		9.3		(4.15)		0.807
Level of spHDM IgE (kU/mL), mean (SD) (Checked randomly by 10% of the total sample)	20.5	(8.75)		20.3		(8.66)		0.685

**Table 2 cells-10-01841-t002:** The effect of SCIT on symptom scores, drug scores, SPT of house-dust-mite diameter, IgE-specific house-dust-mite.

Variable	SCIT Group (*n* = 1.098)		Control Group (*n* = 1.098)		*p*
Symptom Score (SS), mean (SD)					
0–3 months	2.7	(0.48)	2.7	(0.45)	0.390
4–6 months	1.7	(0.53)	2.4	(0.48)	0.000
7–12 months	0.8	(0.61)	2.2	(0.69)	0.000
13–18 months	1.2	(0.39)	2.0	(0.46)	0.000
Difference before-after	1.8	(0.45)	0.4	(0.48)	0.000
Medication Score (MS), mean (SD)					
0–3 months	2.6	(0.48)	2.6	(0.48)	0.860
4–6 months	1.7	(0.53)	2.4	(0.48)	0.000
7–12 months	0.8	(0.61)	2.2	(0.69)	0.000
13–18 months	0.4	(0.59)	2.2	(0.69)	0.000
Difference before-after	2.3	(0.59)	0.4	(0.49)	0.000
Combination Symptom & Medication Score (CSMS), mean (SD)					
0–3 months	5.0	(0.76)	5.0	(0.76)	0.955
4–6 months	3.5	(1.07)	4.7	(0.96)	0.000
7–12 months	1.6	(1.23)	4.4	(1.38)	0.000
13–18 months	1.5	(0.74)	4.2	(0.68)	0.000
Difference before-after	4.1	(0.80)	0.8	(0.84)	0.000
Skin Prick Test Diameter (Mite) (mm), mean (SD)					
before	9.3	(4.17)	9.3	(4.15)	0.807
after	6.2	(1.14)	7.6	(3.91)	0.000
Difference	2.4	(1.26)	1.7	(0.71)	0.000
Level of spHDM IgE (kU/mL), mean (SD) (Checked randomly by 10% of the total sample)					
before	20.5	(8.75)	20.3	(8.66)	0.685
after	12.1	(3.07)	16.4	(9.58)	0.000
Difference	8.4	(8.93)	3.9	(1.73)	0.000

**Table 3 cells-10-01841-t003:** The effect of SCIT on points of symptoms (Nasal, Eye, and Lung Symptoms).

Points of Symptom		Baseline	0–3 Months	4–6 Months	7–12 Months	13–18 Months
Total Points of Symptom, mean (SD)						
SCIT Group	Mean (SD)	16.2 (2.75)	7.2 (1.84)	5.2 (1.60)	2.5 (1.84)	3.5 (1.18)
Non SCIT (Control) Group	Mean (SD)	16.3 (2.70)	7.2 (1.84)	7.1 (1.44)	6.6 (2.07)	6.1 (1.39)
	*p*	0.398	0.944	0.000	0.000	0.000
Nasal Symptom Points of Itchy Nose, mean (SD)						
SCIT Group	Mean (SD)	2.7 (0.44)	1.2 (0.87)	1.6 (0.48)	0.8 (0.56)	1.1 (0.39)
Non SCIT (Control) Group	Mean (SD)	2.8 (0.43)	1.3 (0.78)	1.8 (0.42)	1.7 (0.46)	1.54 (0.88)
	*p*	0.374	0.010	0.000	0.000	0.000
Nasal Symptom Points of Sneezing, mean (SD)						
SCIT Group	Mean (SD)	2.9 (0.28)	1.4 (1.03)	1.7 (0.61)	0.8 (0.58)	1.1 (0.41)
Non SCIT (Control) Group	Mean (SD)	2.9 (0.28)	1.5 (1.00)	2.4 (0.68)	1.9 (0.64)	1.7 (1.01)
	*p*	0.818	0.001	0.000	0.000	0.000
Nasal Symptom Points of Runny Nose (rhinorrhea), mean (SD)						
SCIT Group	Mean (SD)	2.7 (0.44)	1.0 (0.63)	1.6 (0.49)	0.8 (0.55)	1.1 (0.39)
Non SCIT (Control) Group	Mean (SD)	2.8 (0.43)	1.2 (0.67	1.8 (0.43)	1.6 (0.49)	1.5 (0.88)
	*p*	0.374	0.000	0.000	0.000	0.000
Nasal Symptom Points of Blocked Nose (nasal congestion), mean (SD)						
SCIT Group	Mean (SD)	2.6 (0.89)	1.9 (1.43)	0.2 (0.64)	0.1 (0.37)	0.0 (0.36)
Non SCIT (Control) Group	Mean (SD)	2.7 (0.86)	1.9 (1.45)	0.9 (1.17)	1.0 (1.44)	1.1 (1.44)
	*p*	0.508	0.953	0.000	0.000	0.000
Points of Eye Symptom, mean (SD)						
SCIT Group	Mean (SD)	2.7 (0.44)	0.2 (0.55)	0.0 (0.00)	0.0 (0.13)	0.0 (0.00)
Non SCIT (Control) Group	Mean (SD)	2.8 (0.43)	0.2 (0.64)	0.0 (0.00)	0.0 (0.00)	0.0 (0.00)
	*p*	0.374	0.007	0.000	0.000	0.000
Points of Lung Symptom, mean (SD)						
SCIT Group	Mean (SD)	2.1 (1.30)	1.6 (1.48)	0.0 (0.36)	0.0 (0.21)	0.0 (0.18)
Non SCIT (Control) Group	Mean (SD)	2.1 (1.31)	1.1 (1.43)	0.3 (0.94)	0.3(0.94)	0.3 (0.92)
	*p*	0.769	0.000	0.000	0.000	0.000

**Table 4 cells-10-01841-t004:** Frequencies of resources utilization.

Health Care Use		0–3 Months	4–6 Months	7–12 Months	13–18 Months	Total
Physician Visit						
SCIT Group	Mean	9.1	9.1	8.1	3.5	29.8
	SD	(2.69)	(0.50)	(0.44)	(1.18)	(3.34)
Non SCIT (Control) Group	Mean	9.5	8.2	7.6	6.1	31.4
	SD	(2.42)	(1.99)	(1.64)	(1.39)	(7.07)
	*p*	0.220	0.000	0.000	0.000	0.000
Medical Prescription						
SCIT Group	Mean	29.7	15.9	6.7	1.4	53.7
	SD	(10.87)	(5.26)	(4.96)	(3.02)	(17.79)
Non SCIT (Control) Group	Mean	30.0	19.8	17.1	14.0	80.9
	SD	(10.41)	(4.86)	(5.73)	(5.46)	(20.70)
	*p*	0.537	0.000	0.000	0.000	0.000
Hospitalizations						
SCIT Group	Mean	0.00	0.01	0.01	0.00	0.03
	SD	(0.09)	(0.19)	(0.17)	(0.03)	(0.27)
Non SCIT (Control) Group	Mean	0.01	0.01	0.01	0.01	0.04
	SD	(0.16)	(0.19)	(0.19)	(0.30)	(0.43)
	*p*	0.317	1.000	0.811	0.371	0.313

**Table 5 cells-10-01841-t005:** Pharmacotherapy and physiotherapy costs.

Pharmacotherapy (X IDR 1000) Costs		0–3 Months	4–6 Months	7–12 Months	13–18 Months	Total
Antihistamine/decongestant combinations w/or w/o analgesic or cough suppressant (X IDR 1000)						
SCIT Group	Mean	1102	1104	721	135	3062
	SD	(569.4)	(459.4)	(577.8)	(258.2)	(1490.6)
Non SCIT (Control) Group	Mean	1017	880	818	655	3370
	SD	(496.0)	(418.4)	(378.7)	(297.5)	(1566.5)
	*p*	0.000	0.000	0.000	0.000	0.000
Local (nasal) corticosteroids, w/or w/o Inhaled corticosteroids (X IDR 1000)						
SCIT Group	Mean	3975	2774	315	480	7544
	SD	(1973.6)	(2252.8)	(1131.6)	(898.9)	(4504.6)
Non SCIT (Control) Group	Mean	3601	3120	2389	2323	11,433
	SD	(1744.6)	(1480.3)	(1565.4)	(1067.1)	(5268.5)
	*p*	0.220	0.000	0.000	0.000	0.000
Systemic corticosteroids (X IDR 1000)						
SCIT Group	Mean	858	57	15	13	942
	SD	(757.8)	(268.9)	(127.5)	(110.4)	(919.3)
Non SCIT (Control) Group	Mean	820	344	331	292	1787
	SD	(694.6)	(475.1)	(457.8)	(405.4)	(1700.3)
	*p*	0.000	0.000	0.000	0.000	0.000
Inhaled long-acting β2 agonists, w/or w/o Short-acting β2 agonists (X IDR 1000)						
SCIT Group	Mean	254	6	1	1	262
	SD	(269.4)	(50.8)	(20.4)	(17.5)	(284.4)
Non SCIT (Control) Group	Mean	166	37	37	32	272
	SD	(237.0)	(107.7)	(105.0)	(92.2)	(444.8)
	*p*	0.160	0.001	0.000	0.000	0.379
Skin Care (X IDR 1000)						
SCIT Group	Mean	123	3	2	1	129
	SD	(431.5)	(67.6)	(57.5)	(49.3)	(487.0)
Non SCIT (Control) Group	Mean	98	19	18	16	151
	SD	(374.3)	(143.3)	(136.9)	(122.6)	(652.4)
	*p*	0.000	0.000	0.000	0.000	0.000
Physiotherapy (X IDR 1000)						
SCIT Group	Mean	85	5	5	4	99
	SD	(822.3)	(163.7)	(163.7)	(140.3)	(1033.3)
Non SCIT (Control) Group	Mean	748	63	59	53	923
	SD	(2373.4)	(563.4)	(524.6)	(478.3)	(3137.0)
	*p*	0.000	0.001	0.001	0.001	0.000
Total Pharmaco-physiotherapy (X IDR 1000)						
SCIT Group	Mean	6396	3948	1058	635	12,037
	SD	(3103.4)	(2561.3)	(1490.9)	(1245.3)	(6699.0)
Non SCIT (Control) Group	Mean	6450	4464	3651	3371	17,935
	SD	(3483.7)	(1945.5)	(2024.5)	(1538.9)	(7661.5)
	*p*	0.703	0.000	0.000	0.000	0.000

**Table 6 cells-10-01841-t006:** Health care costs.

Health Care Costs			0–3 Months	4–6 Months	7–12 Months	13–18 Months	Total
Medical Care Costs							
Physician Costs (X IDR 1000)	SCIT Group	Mean	1163	1161	1035	443	3802
		SD	(423.6)	(236.3)	(210.6)	(177.1)	(880.8)
	Non SCIT (Control) Group	Mean	1228	1064	989	793	4073
		SD	(399.1)	(334.0)	(289.5)	(236.4)	(1219.5)
		*p*	0.000	0.000	0.000	0.000	0.000
Laboratory Costs (X IDR 1000)	SCIT Group	Mean	2933			3569	6502
		SD	(55.4)			(156.7)	(186.6)
	Non SCIT (Control) Group	Mean	2928			3564	6491
		SD	(64.5)			(163.7)	(204.3)
		*p*	0.076			0.000	0.000
Hospitalization Cost (X IDR 1000)	SCIT Group	Mean	3	16	14	1	33
		SD	(108.6)	(219.2)	(195.1)	(31.7)	(313.4)
	Non SCIT (Control) Group	Mean	9	16	15	10	49
		SD	(164.5)	(217.4)	(206.5)	(316.9)	(464.4)
		*p*	0.371	1.000	0.861	0.371	0.361
**Total Pharmaco-physiotherapy Costs** (X IDR 1000)	SCIT Group	Mean	6396	3948	1058	635	12,037
		SD	(3103.4)	(2561.3)	(1490.9)	(1245.3)	(6699.0)
	Non SCIT (Control) Group	Mean	6450	4464	3651	3371	17,935
		SD	(3483.7)	(1945.5)	(2024.5)	(1538.9)	(7661.5)
		*p*	0.703	0.000	0.000	0.000	0.000
SCIT Costs (X IDR 1000)	SCIT Group	Mean	2100	600	1200	300	4200
		SD	(0.0)	(0.0)	(0.0)	(0.0)	(0.0)
Total Medical Care Cost (X IDR 1000)	SCIT Group	Mean	12,595	5724	3307	4948	26,575
		SD	(3340.1)	(2582.5)	(1519.2)	(1301.4)	(6926.9)
	Non SCIT (Control) Group	Mean	10,614	5543	4655	7737	28,549
		SD	(3680.3)	(2076.8)	(2054.4)	(1660.5)	(8100.0)
		*p*	0.000	0.070	0.000	0.000	0.000
**Travel Costs**							
Travel Costs (X IDR 1000)	SCIT Group	Mean	3948	3424	3181	2557	13,111
		SD	(1187.4)	(991.9)	(848.0)	(709.1)	(3601.2)
	Non SCIT (Control) Group	Mean	3844	3844	3428	1464	12,581
		SD	(1257.5)	(541.1)	(480.9)	(529.8)	(2165.8)
		*p*	0.047	0.000	0.000	0.000	0.000
**Total Health Care Costs**							
Total Health Care Cost (X IDR 1000)	SCIT Group	Mean	16,439	9568	6735	6413	39,156
		SD	(4204.3)	(2633.8)	(1557.1)	(1549.6)	(7671.3)
	Non SCIT (Control) Group	Mean	14,562	8967	7836	10,294	41,659
		SD	(4394.9)	(2619.7)	(2291.9)	(2054.2)	(10,031.9)
		*p*	0.000	0.000	0.000	0.000	0.000

## Data Availability

The data that support the findings of this study are available from the corresponding author upon reasonable request.

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
