# Peer review of "Indonesia-Based Study of the Clinical and Cost-Saving Benefits of Subcutaneous Allergen Immunotherapy for Children with Allergic Rhinitis in Private Practice"

_cells, 2021, doi:10.3390/cells10071841_

Round 1

Reviewer 1 Report

I was really delighted to read this interesting study on  the clinical and cost-saving benefits of subcutaneous allergen immunotherapy for children with AR. The paper is well structured, heavily up-dated, and using appropriate statistics provides the reader with an accurate and up-to-date point of view regarding the importance of subcutaneous allergen immunotherapy in the treatment of children with AR. It is a must for every clinician and health care provider involved in the treatment of children with AR.

Author Response

Thank you!

Reviewer 2 Report

【General comments】

This is a study to evaluate the the clinical and cost-saving benefits of subcutaneous allergen immunotherapy for children with allergic rhinitis in private practice.

This is a well-designed and well-described manuscript.

However there are some points that must be cleared out.

【Specific comments】

(1)Table 1

As for inclusion criteria, severity of symptom should be normalized as the same ARIA classification. Please inform whether there existed any symptom severity difference between two groups before study enrollment

(2) Table 2

In Table2, the effect of SLIT on symptom scores(total) is showed.

Can authors show any data about the effect on nasal symptom scores(sneezing, rhinorrhea, and nasal congestion) ?

Please let me have a comment .

I hope that my comments are useful for the improvement of this manuscript.

Author Response

We agree with the reviewer’s assessment. Accordingly, throughout the manuscript, we have answered some points

1. (Table 1)

We have added the baseline symptom score, medication score, combination before study enrolment. There are no difference between groups in the symptoms score (base of ARIA classification) before study enrolment.

2. (Table 2)

Yes, we can. We have added the data regarding nasal symptom points of score data. Data about the effect on nasal symptom scores (sneezing, rhinorrhea, and nasal congestion) were added to a new table (Table 3).

Reviewer 3 Report

The article is interesting and well written. The argument is not new but the cut linked to the economic aspects in a particular Nation makes it unique. The study has the limits of a retrospective study but the large number of sample population ensures the validity of the results. English is correct and reading is fluent. I approve its publication after specifying the adverse reactions of the drug.

Author Response

Thank you for your constructive comment and suggestion. Some of the most common adverse reactions associated with immunotherapy treatment is chills, constipation, coughing, decreased appetite, diarrhea, fatigue, fever and flu-like symptoms, headache, infusion-related reaction or injection site pain, itching, localized rashes and/or blisters.